# Temporal Memory Enhancement for Semantic Segmentation in Surgical Video

**Zheyao Gao**[1]        ZHEYAOGAO@CUHK.EDU.HK
**Qian Wu**[1]        QIANWU@LINK.CUHK.EDU.HK
**Yueyao Chen**[1]        CHENYUEYAO@LINK.CUHK.EDU.HK
**Cheng Chen**[2]        CCHEN@EEE.HKU.HK
**Hon Chi Yip**[3]        HCYIP@SURGERY.CUHK.EDU.HK
**Winnie Chiu Wing Chu**[4]        WINNIECHU@CUHK.EDU.HK
**Qi Dou**[1]        QIDOU@CUHK.EDU.HK

[1] *Department of Computer Science and Engineering, The Chinese University of Hong Kong*

[2] *Department of Electrical and Electronic Engineering, Hong Kong University*

[3] *Department of Surgery, The Chinese University of Hong Kong*

[4] *Department of Imaging and Interventional Radiology, The Chinese University of Hong Kong*

**Editors:** Accepted for publication at MIDL 2026

## Abstract

Segmenting critical anatomical structures in surgical videos can enhance precision and patient safety by alerting surgeons to potential complications. While current methods that store features from past frames have advanced the performance in video segmentation, their reliance on a fixed-range local memory often fails to capture complex temporal contexts of surgical scenes. Specifically, the memory could fill with redundant features or omit informative frames due to the non-uniform rate of operations by the surgeons. Besides, the image features in the same phase of the surgery share similar patterns, while local memory could not capture such long-term relationships. Therefore, we propose a memory enhancement method to enrich the local temporal context and incorporate global phase context for surgical video semantic segmentation. Concretely, we improve the local memory with a feature selection module based on Determinantal Point Process (DPP) to choose past features that are diverse and relevant to the current feature. Besides, we introduce a global memory to store the common patterns of frames within each phase based on the conditional variational autoencoder with a mixture of Gaussian priors (CVAE-MoG). Experiments on endoscopic submucosal dissection (ESD) and laparoscopic cholecystectomy (LC) video segmentation demonstrate that our method achieves superior performance over existing methods. Our code is available via https://github.com/key1589745/surgery_segmentation.

**Keywords:** Surgery, video segmentation, temporal memory.

## 1. Introduction

The analysis of surgical video, such as from endoscopic submucosal dissection (ESD) or laparoscopic cholecystectomy (LC), is essential for developing intraoperative guidance systems. In ESD, segmenting structures like the submucosa is key for safe lesion removal (Yang et al., 2023), while in LC, identifying critical anatomy like the cystic duct is paramount for preventing complications (Murali et al., 2023). However, a fundamental challenge across

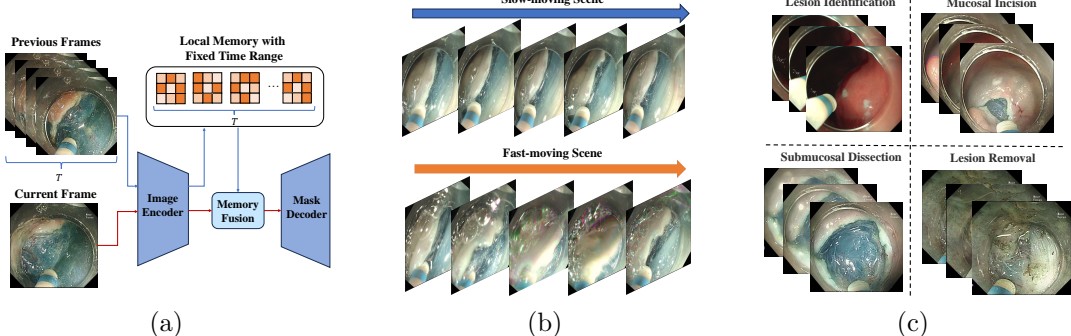

Figure 1: The typical memory-based video segmentation framework and its limitations in surgical video segmentation. (a) Video segmentation methods such as SAM2 (Ravi et al., 2024) apply a local memory bank to store the features of the past few frames within a fixed time interval. (b) The temporal dynamics vary in surgical videos. (c) Frames within each phase of the surgery share similar patterns.

these domains is the high variability of surgical video—characterized by changes in camera motion, lighting, and tissue appearance—which often causes segmentation and detection models to produce inconsistent, frame-by-frame predictions (Mascagni et al., 2025).

Different from image segmentation, a crucial problem in video segmentation lies in how to leverage temporal information effectively to delineate regions of interest in current frames. To fully exploit the temporal dependency between frames, previous works (Wang et al., 2021; Zhuang et al., 2024, 2022) have proposed various feature extraction and memory mechanisms to capture the temporal relations and integrate the information from the past with current frames over the video sequence. Fig. 1(a) shows a typical framework of video segmentation with a memory module, which incorporates temporal context by storing features of past frames and their segmentation masks as references. Recently, SAM2 (Ravi et al., 2024) and its variants (Liu et al., 2024) have leveraged the progress of such memory strategies to develop foundation models for video segmentation, which have achieved excellent performance in both natural and surgical scenes.

While video segmentation methods commonly employ a local memory bank to store features of recent frames, this can be insufficient for capturing long-term and complex temporal dependencies. To address this, advanced memory-based frameworks (Cheng and Schwing, 2022; Deng et al., 2024) incorporate both a local working memory and a long-term memory component, thereby extending the model's effective temporal receptive field to hundreds or even thousands of frames. However, these existing methods remain suboptimal for endoscopic video segmentation due to two distinctive properties of surgical workflows. First, as illustrated in Fig. 1(b), the surgeon's motion pace is highly variable. A static memory update strategy may store excessive redundant features during slow, deliberate phases while missing critical reference frames during periods of rapid movement. Second, surgical procedures follow a structured workflow. As shown in Fig. 1(c), frames within the same surgical phase share similar visual patterns—such as tissue appearance and lighting condi-

tions—that constitute a long-term, phase-specific context. Current memory mechanisms, which primarily rely on feature similarity or recency, fail to explicitly model and leverage this high-level semantic consistency across the entire procedure.

To address the complex temporal context in surgical video segmentation, we propose a memory enhancement strategy comprising two key mechanisms. First, to reduce memory redundancy while preserving valuable past features, we design a feature selection module based on a greedy Determinantal Point Process (DPP) (Anari et al., 2025). This module efficiently selects a diverse and relevant subset from local memory to enhance the current feature representation. Second, to capture global phase context, we propose a conditional variational autoencoder with a mixture of Gaussian prior (CVAE-MoG) that encodes common patterns of each surgical phase. This module clusters frames by phase and integrates phase-specific context from the corresponding cluster to enhance current features.

Our contributions include: (1) we propose a temporal memory enhancement method for semantic segmentation in surgical video, which improves the segmentation performance by incorporating richer local context and long-term phase context; (2) we propose a local memory feature selection module based on a greedy DPP to address the varying movement rates in surgical videos. It selects diverse and relevant features from memory while filtering out redundancies; (3) we design a global memory module based on CVAE-MoG, which exploits common patterns of frames in each phase and provides the long-term phase context to enhance the segmentation performance; (4) we validated the effectiveness of the proposed method on two surgical video segmentation datasets. The results show that our method achieves superior performance over existing video segmentation methods.

## 2. Related Works

### 2.1. Video Semantic Segmentation

The core challenge in Video Semantic Segmentation (VSS) is maintaining temporal coherence to ensure consistent pixel classification across frames. Traditional approaches, such as optical flow-based propagation (Nilsson and Sminchisescu, 2018), 3D convolutional networks (He et al., 2017), and recurrent or attention-based mechanisms (Li et al., 2025), have faced challenges in generalizing to variable scenes while balancing robust temporal consistency with computational efficiency. Recently, the Segment Anything Model 2 (SAM2) (Ravi et al., 2024) has emerged as a promising foundation model. Pre-trained on large-scale datasets, SAM2 enables promptable video object segmentation and has demonstrated strong generalization across diverse scenes (Zhao et al., 2025). This capability has prompted recent research (Syed Ariff et al., 2025) to adapt SAM2 for the VSS task, offering a new pathway toward achieving coherent, efficient, and generalizable video segmentation.

### 2.2. Surgical Scene Segmentation

Recent research in surgical video segmentation has rapidly evolved, moving from developing specialized models (Jin et al., 2022; Wang et al., 2024; Li et al., 2023) to adapting powerful foundation models like SAM2 (Ravi et al., 2024) for the surgical domain. Adaptations such as SurgiSAM2 demonstrate that fine-tuning the foundational model on multi-dataset surgical annotations significantly boosts anatomy segmentation performance (Devanish et al.,

[2025](#)); Surgical SAM 2 introduces an efficient frame pruning mechanism, reducing computational cost ([Liu et al., 2024](#)); SAM2S establishes a new benchmark by integrating semantic learning and a diverse memory mechanism for long-term, class-aware instrument and tissue tracking ([Liu et al., 2025a](#)). While performance on larger structures such as surgical tools has become more robust, significant challenges remain in achieving reliable segmentation of small, critical structures like nerves and blood vessels ([Wu et al., 2024](#)).

### 2.3. Temporal Memory enhancement

Memory-based methods improve video segmentation by storing and retrieving past frame features, ensuring temporal consistency on various time scales. For long-term consistency, models like STMN ([Oh et al., 2019](#)) use an external memory bank, while XMem ([Cheng and Schwing, 2022](#)) adopts a multi-store architecture to retain information across thousands of frames. Specialized tools such as Memsam ([Deng et al., 2024](#)) further adapt this for domains like echocardiography analysis. For local refinement, methods enhance the robustness and quality of the memory: SAMURAI ([Yang et al., 2024](#)) adds context-aware and occlusion-resilient memory to SAM2, and MA-SAM2 ([Yin et al., 2025](#)) adapts it for surgical videos; QDMN ([Liu et al., 2025b](#)) employs a dynamic memory bank that stores only high-quality masks to reduce error propagation.

### 3. Method

The aim of the proposed method is to improve the latent feature $f$ learned by the image encoder of a segmentation network by incorporating both local and global phase context. Fig. 2 shows our framework.

During training, the image encoder extracts memory features $\{f_{-t}\}_{t=1}^{T}$ from past frames $\{x_{-t}\}_{t=1}^{T}$ and the current feature $f$ from the frame $x$. The local memory evaluates similarities among memory features and with the current feature, selecting a subset $\mathcal{B}$ that is both diverse and relevant as local context. The global memory concatenates the enhanced feature $f'$ with phase context $\tilde{z}^l$, which is sampled from $q(z, c|x, y)$ estimated by post network $\beta(\cdot)$. The training objective combines segmentation loss $\ell_{CE}$ with KL divergence loss $\ell_{KL}$, which aligns $p(z, c|x, y)$ estimated by the prior network $\alpha(\cdot)$ with $q(z, c|x, y)$.

During inference, the local memory only evaluates similarities between the current feature and memory features and stores the similarities for subset selection. The global memory generates $\tilde{z}^l$ from $p(z, c|x)$ since the ground truth mask is not available.

We elaborate the details of the local memory and global memory in the following sections.

### 3.1. Local Memory with DPP selection

Given the current frame feature $f \in \mathbb{R}^C$ and memory cache: $\mathcal{A} = \{f_{-t}\}_{t=1}^{T}$ where $f_{-t} \in \mathbb{R}^C$, we apply the DPP algorithm to choose $M$ features $\mathcal{B} = \{f_{-t}\}_{t=1}^{M}$ where $M < T$. The selection should maximize relevance - features similar to the current frame $f$ and diversity - selected features are different from each other.

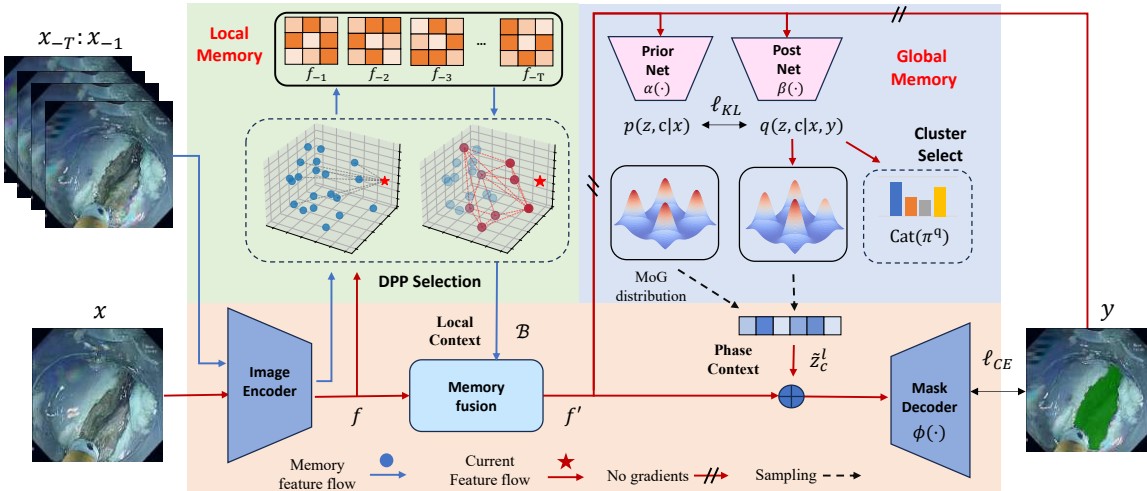

Figure 2: The framework of temporal memory enhancement method for video semantic segmentation. The local memory collects features of past $T$ frames and applies greedy Determinantal Point Process (DPP) to select diverse and relevant features to enhance the current feature. The global memory models the common image and segmentation pattern within each phase as a Mixture of Gaussian (MoG) distribution. For each frame, it selects the most probable component and samples a phase context $\tilde{z}_c^l$, which is then concatenated with the feature $f'$.

To prepare for the selection, the similarity matrix $S \in \mathbb{R}^{T \times T}$ and the relevance matrix $R \in \mathbb{R}^{T \times T}$ are calculated based on cosine similarity as follows,

$$S_{\mathcal{A}} = \bar{f}_{\mathcal{A}} \bar{f}_{\mathcal{A}}^{\top}, \quad R_{\mathcal{A}} = \mathrm{diag}\left(\exp(\gamma \cdot \bar{f}_{\mathcal{A}} \bar{f}^{\top})\right), \tag{1}$$

where $\bar{f}_{\mathcal{A}} = [\frac{f_{-1}}{\|f_{-1}\|_2}, \frac{f_{-2}}{\|f_{-2}\|_2}, ..., \frac{f_{-T}}{\|f_{-T}\|_2}] \in \mathbb{R}^{T \times C}$ and $\bar{f} = \frac{f}{\|f\|_2} \in \mathbb{R}^C$ are normalized memory and current features, respectively; $\gamma \in [0, 1]$ is a hyperparameter, balancing the weight of the relevance score in the selection; $\mathrm{diag}(\cdot)$ denotes the diagonal matrix formed by a vector. For streaming data, the cosine similarity $\bar{f}_{\mathcal{A}} \bar{f}^{\top}$ is calculated and cached in the memory during inference. The similarity matrix $S$ could be derived by reusing the cached similarities, avoiding additional computation.

To find the optimal subset $\mathcal{B}^*$, we define the probability distribution $p(\mathcal{B})$ over all possible subsets of $\mathcal{A}$ as follows,

$$p(\mathcal{B}) \propto \det(R_{\mathcal{B}} S_{\mathcal{B}} R_{\mathcal{B}}) = \left(\prod_{t \in \mathcal{B}} r_t\right)^2 \cdot \det(S_{\mathcal{B}}), \tag{2}$$

where $r_t = \exp(\gamma \cdot \bar{f}_t \bar{f}^{\top})$ denotes a diagonal element in $R_{\mathcal{B}}$ and $\det(\cdot)$ denotes the determinant of a matrix. It could be observed based on Eq. (1) and (2) that the probability of selecting

a subset correlates positively with similarities to the current frame (*i.e.*, diagonal elements of $R_{\mathcal{B}}S_{\mathcal{B}}R_{\mathcal{B}}$) and correlates negatively with the similarities between memory features (*i.e.*, off-diagonal elements). Therefore, the subset that contains diverse and relevant features could be derived through Maximum A Posteriori (MAP) inference,

$$\mathcal{B}^* = \arg\max_{\mathcal{B}\subseteq\mathcal{A},|\mathcal{B}|=M} p(\mathcal{B}). \tag{3}$$

Finding the exact MAP (the subset with the absolute highest probability) by listing all combinations is compute-intensive. Therefore, we apply a greedy update algorithm (Gautier et al., 2019), which builds the subset one item at a time and always picks the memory feature maximizing the determinant increase. It reduces the time complexity to $o(TM^2)$, which is efficient when the size of working memory is small. The details of this algorithm are provided in Appendix A.

### 3.2. Global Memory based on CVAE-MoG

To incorporate the long-term memory information in the current feature $f'$ from the preceding stage, we propose to represent the common pattern of the image $x$ and segmentation $y$ of each phase with the latent phase context $z$ that comes from a MoG distribution $\sum_{c=1}^{C}\pi_c\mathcal{N}(\mu_c,\sigma_c^2 I)$, where $\mu_c,\sigma_c^2 \in \mathbb{R}^d$ are the parameters of the $c$th Gaussian component in the mixture distribution conditioned on the image and segmentation. $\pi_c \in [0,1]$ is the mixing probability.

The MoG distribution is estimated based on the idea of conditional variational autoencoder (Kohl et al., 2018). Concretely, the objective is to find a variational distribution $q(z,c|x,y)$ to approach the true posterior by maximizing the lower bound of the likelihood $p(y|x)$ written as follows,

$$\begin{aligned}
\log p(y|x) &= \log\int\sum_{c=1}^{C}p(y,z,c|x)\,dz \\
&\geq \mathbb{E}_{q(z,c|x,y)}[\log p(y|z,x)] - \mathcal{D}_{KL}[q(z,c|x,y)||p(z,c|x)],
\end{aligned} \tag{4}$$

where $\mathcal{D}_{KL}$ is the KL divergence. Since the lower bound could not be calculated explicitly, we introduce the prior network $\alpha(\cdot)$ and posterior network $\beta(\cdot)$ to estimate the parameters of $p(z,c|x)$ and $q(z,c|x,y)$, respectively, and maximize the lower bound through gradient descent. Formally, the distributions above are derived based on the following definitions,

$$p(z|c,x) = \mathcal{N}(\mu_c(x;\alpha),\sigma_c^2(x;\alpha)I), \qquad p(c|x) = \mathrm{Cat}(\pi^p(x;\alpha)), \tag{5a}$$

$$q(z|x,y) = \mathcal{N}(\mu(x,y;\beta),\sigma^2(x,y;\beta)I), \qquad q(c|x,y) = \mathrm{Cat}(\pi^q), \tag{5b}$$

where the mean field assumption is used in variational approximation, *i.e.*, $q(z,c|x,y) = q(z|x,y)q(c|x,y)$. Following the previous work (Jiang et al., 2017), the mixing probability $\pi^q$ of the variational distribution is calculated as follows,

$$\pi_c^q = \mathbb{E}_{q(z|x,y)}[q(c|z)] \approx \frac{q(z=\tilde{z}^l|c,x,y)p(c)}{\sum_{c'=1}^{C}q(z=\tilde{z}^l|c',x,y)p(c')}, \tag{6}$$

where $\tilde{z}^l \sim q(z|x, y)$ and $p(c)$ is set to $\frac{1}{C}$ for each cluster. Based on the derivation of the above distributions, the first term in Eq. (4) could be maximized by minimizing the cross-entropy loss between the prediction of the mask decoder $\phi(\cdot)$ and the ground truth $\hat{y}$. Formally, the loss function for this term could be written as follows,

$$\ell_{seg} = \text{CE}(\phi(f' \oplus \tilde{z}_c^l), \hat{y}), \tag{7}$$

where $\tilde{z}_c^l$ is sampled from $q(z|x, y)$ using the reparameterization trick (Kingma et al., 2013) such that the posterior encoder could be updated by backpropagation; $\oplus$ denotes contention in the channel dimension by broadcasting $\tilde{z}_c^l$ along the spatial dimension of $f'$.

The KL term could be derived explicitly by using the definitions in Eq. (5a) and (5b). Thus, this term could be minimized as follows,

$$\ell_{KL} = \sum_{c=1}^{C} \pi_c^q \left( \log \frac{\sigma_c(x; \alpha)}{\sigma_c(x, y; \beta)} + \frac{\sigma_c^2(x; \alpha) + \left( \mu_c(x; \alpha) - \mu_c(x, y; \beta) \right)^2}{2\sigma_c^2(x, y; \beta)} - \log \pi_c^p(x; \alpha) \right). \tag{8}$$

$\ell_{KL}$ forces $p(z, c|x)$ to be close with $q(z, c|x, y)$ that encodes the pattern of the segmentation mask. Therefore, the samples from $p(z|c, x)$ could be used as the phase context during inference when ground truth is not available. The detailed derivation for the lower bound and KL loss is provided in Appendix B.

## 4. Experiments

We validate our method on two surgical video datasets: ESD and LC. In Section 4.2, we compare it against state-of-the-art video segmentation methods, followed by an ablation study and hyperparameter analysis in Section 4.3.

### 4.1. Dataset and Implementation details

**Dataset.** ESD dataset consisted of 1,201 annotated frames from 34 endoscopic surgical videos. For each video, submucosal, muscular tissue, and blood vessels in critical frames were annotated by expert surgeons. The dataset is available from the corresponding author upon reasonable request for non-commercial purposes. LC dataset comes from the public dataset Endoscapes-Seg50 (Murali et al., 2023), containing 50 laparoscopic cholecystectomy videos, of which 493 frames are annotated with instance and semantic segmentation masks for six classes (five anatomical structures and one tool). Annotations are provided sparsely at a rate of one frame every 30 seconds, with splits of 30 training, 10 validation, and 10 test videos. For pre-processing, all the images were normalized to the same mean and variance as the training data in the previous work (Ravi et al., 2024) and resized to $256 \times 256$ as the network inputs.

**Implementation details.** We used the Hiera-large encoder pretrained by SAM2 (Ravi et al., 2024) as the backbone of the image encoder. Similar to the previous work (Chen et al., 2023), we froze most of the parameters in the encoder and only trained the adapters. In local memory, the memory size $T$ and the subset size $M$ were set to 16 and 8 for the ESD dataset, 8 and 4 for the LC dataset. The time intervals between past frames were set to 0.5 and 1 based on the experience of previous works (Cao et al., 2023; Murali et al., 2023) for

Table 1: Comparison results with state-of-the-art video segmentation methods.

| Methods | LC dataset | | | ESD dataset | | | FPS |
|---|---|---|---|---|---|---|---|
| | mIoU (%) | F-score (%) | wDice (%) | mIoU (%) | F-score (%) | wDice (%) | |
| Mask2Former | $83.7 \pm 1.89$ | $39.6 \pm 4.07$ | $86.9 \pm 1.43$ | $81.9 \pm 2.45$ | $72.2 \pm 2.67$ | $85.8 \pm 1.76$ | 82.7 |
| FRGM | $82.9 \pm 1.78$ | $39.2 \pm 3.17$ | $86.2 \pm 1.52$ | $81.1 \pm 2.55$ | $71.5 \pm 2.38$ | $85.0 \pm 1.92$ | 111.9 |
| QDMN++ | $84.5 \pm 1.55$ | $42.9 \pm 3.71$ | $89.0 \pm 0.94$ | $83.4 \pm 2.07$ | $72.8 \pm 2.59$ | $87.1 \pm 1.28$ | 78.4 |
| MemSAM | $84.9 \pm 1.38$ | $44.5 \pm 3.53$ | $89.7 \pm 0.70$ | $84.1 \pm 1.88$ | $73.1 \pm 2.55$ | $87.8 \pm 1.04$ | 57.8 |
| SAM2 | $84.7 \pm 1.32$ | $46.1 \pm 3.48$ | $90.8 \pm 3.82$ | $85.1 \pm 1.81$ | $73.6 \pm 2.84$ | $88.2 \pm 0.78$ | 48.5 |
| SurgSAM2 | $85.1 \pm 1.21$ | $45.6 \pm 3.35$ | $91.0 \pm 0.45$ | $84.8 \pm 1.69$ | $73.3 \pm 2.51$ | $88.4 \pm 0.79$ | 76.1 |
| Ours | $\mathbf{86.6 \pm 0.83}$ | $\mathbf{47.8 \pm 3.21}$ | $\mathbf{92.7 \pm 0.38}$ | $\mathbf{86.9 \pm 1.13}$ | $\mathbf{76.4 \pm 2.12}$ | $\mathbf{90.5 \pm 0.59}$ | 66.3 |

Figure 3: Per-class F-score and Dice score on the LC dataset (a-b) and ESD dataset (c-d).

the ESD dataset and LC dataset, respectively. For the memory fusion module, we used the two layers of cross-attention as used in SAM2. In terms of the prior and posterior networks, we applied a similar configuration as in the previous work (Kohl et al., 2018) and set the number of components of the MoG as 4 for the ESD dataset and 3 for the LC dataset in the comparison study. For the mask decoder, we applied a lightweight decoder network designed in the SegFormer (Xie et al., 2021) model. The framework was trained using the Adam optimizer with an initial learning rate of $6e - 4$ for 300 epochs, which decayed by a factor of 0.1 on the plateau of validation loss. The whole framework was implemented with PyTorch on an RTX 4090 GPU.

**Experiment settings.** For experiments on the ESD dataset, we randomly split the videos into proportions of 60%, 20%, and 20% for training, validation, and testing, respectively. For experiments on the LC dataset, we follow the split of training, validation, and testing in the previous work (Murali et al., 2023). The evaluation metrics include the average of Intersection over Union (mIoU), the boundary F-score (Wang et al., 2022), and the weighted average of Dice score (wDice) according to the occurrence of each structure (Sudre et al., 2017). Please refer to Appendix C for details of the evaluation metrics. The results in all experiments were obtained by performing three independent runs.

## 4.2. Comparison Study

To demonstrate the effectiveness of the proposed memory enhancement method, we compared with widely used methods for video segmentation in natural scene: Mask2Former (Cheng

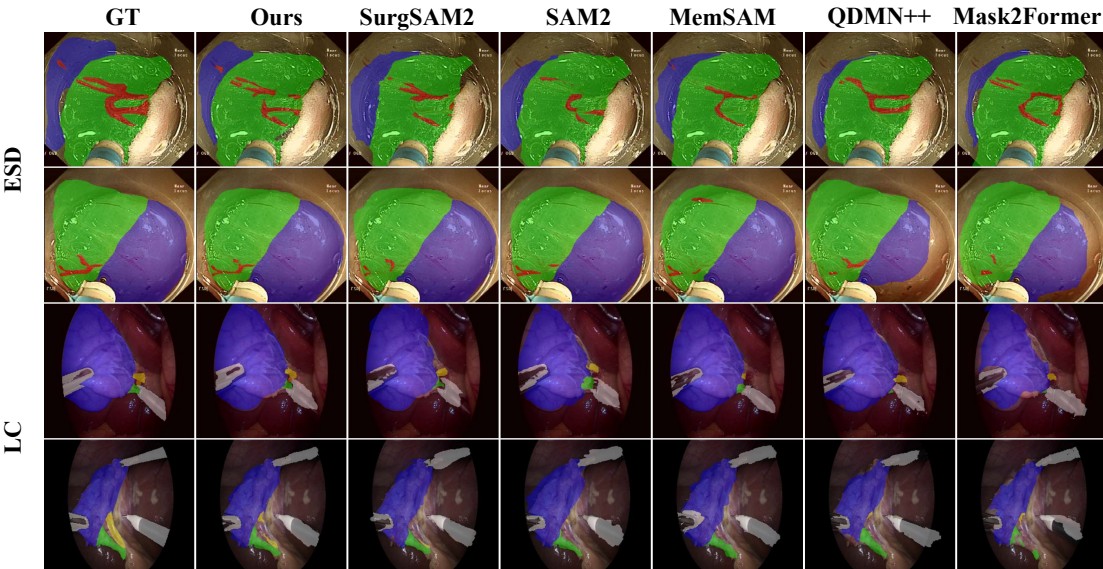

Figure 4: The segmentation results achieved by the proposed method and compared methods on ESD dataset and LC dataset. ESD dataset includes submucosal tissue (green), muscular tissue (blue), and blood vessel (red) and LC dataset includes gallbladder (blue), cystic duct (yellow), cystic artery (green), calot's triangle (orange), and tool (white).

et al., 2022) and SAM2 (Ravi et al., 2024), memory enhancement methods: MemSAM (Deng et al., 2024) that introduces a noise-resilient spatial temporal methods based on SAM, QDMN++ (Liu et al., 2025b) that introduces a quality control module filter low-quality memory features, surgical video segmentation method: FRGM (Yang et al., 2023) which designs a calibration module based on an image segmentation model and SurgSAM2 (Liu et al., 2024), an adaptation of SAM2 to surgical videos. To ensure fair comparison, all methods built upon the SAM architecture—namely, SAM2, MemSAM, SurgSAM2, and our method—use the identical Hiera-Large encoder pre-trained on the SAM2 dataset. They were also fine-tuned using the same adapter-based protocol (Chen et al., 2023) on our surgical video data.

Table 1 presents the comparison results. It could be observed that our proposed method achieves state-of-the-art performance across all segmentation accuracy metrics (mIoU, F-score, wDice) on both the LC and ESD datasets. Among the compared methods, SAM2 and memory-enhanced methods (MemSAM, QDMN++) generally outperform the generic video segmentation model Mask2Former and the surgical-specific FRGM in terms of accuracy, underscoring the value of dedicated memory mechanisms and powerful foundation models. While our method does not achieve the highest FPS, its 66.3 FPS is sufficient for real-time processing and represents a favorable balance, being substantially faster than SAM2 and MemSAM while significantly outperforming them in accuracy. This result confirms

Table 2: Ablation study of the local and global memory.

| Methods | LC dataset | | | ESD dataset | | | FPS |
|---|---|---|---|---|---|---|---|
| | mIoU (%) | F-score (%) | wDice (%) | mIoU (%) | F-score (%) | wDice (%) | |
| Random | 83.5±1.20 | 43.1±3.60 | 88.3±1.10 | 84.5±1.90 | 72.8±2.35 | 87.4±1.24 | 78.1 |
| w/o Relevance | 83.7±1.05 | 44.4±3.43 | 88.2±1.85 | 82.3±2.70 | 73.6±2.15 | 87.9±0.95 | 67.4 |
| w/o Diversity | 84.8±1.80 | 45.2±3.15 | 90.7±1.30 | 86.1±2.35 | 74.9±1.90 | 88.7±0.65 | 72.9 |
| w/o Cluster | 84.2±1.12 | 43.9±2.95 | 89.1±0.92 | 83.4±1.32 | 73.8±2.05 | 87.4±0.88 | 68.5 |
| w/o Mask | 84.8±1.06 | 45.7±2.80 | 89.9±0.85 | 85.8±1.26 | 75.3±1.95 | 88.4±0.80 | 66.7 |
| KM-Cluster | 85.4±0.99 | 46.1±2.65 | 91.5±0.78 | 85.7±1.20 | 75.8±1.85 | 88.6±0.74 | 66.4 |
| Soft-Mix | 86.2±0.85 | 47.6±2.91 | 92.2±0.78 | 86.2±1.12 | 75.2±1.87 | 89.5±0.54 | 66.2 |
| Ours | **86.6 ± 0.83** | **47.8 ± 3.21** | **92.7 ± 0.38** | **86.9 ± 1.13** | **76.4 ± 2.12** | **90.5 ± 0.59** | 66.3 |

that our memory enhancement method effectively leverages temporal information to boost segmentation precision and stability without sacrificing practical applicability.

To provide a more granular assessment of segmentation performance, we have incorporated per-class Dice scores and F-scores in our analysis. These metrics are especially valuable for isolating performance on small, critical anatomical structures—such as the cystic duct and blood vessels—where class imbalance and precise localization are paramount for downstream clinical tasks, including critical view of safety identification and procedural guidance. The detailed results, included in Fig. 3, show that while the performance benefits on larger structures like the gallbladder or submucosal tissue are limited, our approach achieves significant improvement on critical, small-scale anatomy compared to prior video segmentation baselines. This demonstrates that our memory enhancement strategy offers particular practical value where precise segmentation is most safety-critical.

### 4.3. Ablation Study

To validate the design of our proposed local and global memory modules, we compared them with different variants. Concretely, we first studied the importance of the similarity scores and relevance scores in the DPP selection by comparing the proposed selection strategy with the ablated methods (w/o Diversity, w/o Relevance). Besides, the method that applies random selection (Random) was also compared as a baseline. For the global memory, we implemented the method without phase context (w/o Cluster), the method that does not use segmentation masks to generate phase context (w/o Mask), and the method that uses K-means clustering to categorize the features into different clusters (KM-Cluster). Furthermore, we implemented an alternative method (Soft-Mix) that generates the global context $\tilde{z}^l$ by a weighted average of samples from each Gaussian component (*i.e.*, $\tilde{z}^l = \sum_{c=1}^{C} \pi_c \tilde{z}_c^l$) to introduce phase perturbation.

Table 2 presents the comparison results. For local memory, our method—which uses DPP selection with both relevance and diversity scores—achieves the best performance across all primary metrics (mIoU, F-score, wDice) on both datasets. It significantly outperforms the random baseline (e.g., +3.1% mIoU on LC) and the ablated variants ("w/o Relevance" and "w/o Diversity"), confirming that both components are essential. For global memory, the proposed CVAE-MoG clustering method surpasses simplified variants ("w/o Cluster", "w/o Mask", "KM-Cluster"), demonstrating the value of using segmentation masks to create discriminative phase contexts. While the full method has a slightly

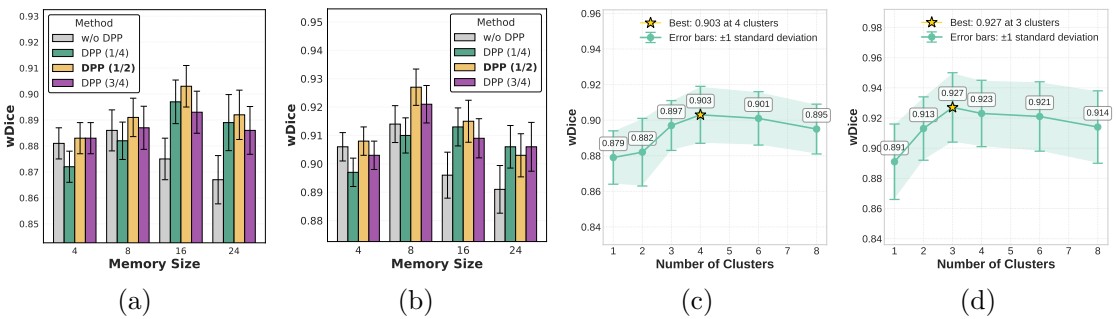

Figure 5: Analysis of the memory size on ESD dataset (a) and LC dataset (b) and the cluster numbers on ESD dataset (c) and LC dataset (d).

lower FPS (66.3) than Random (78.1), this trade-off is justified by its substantially superior segmentation accuracy and robustness. In comparison with Soft-Mix, our method achieved superior performance in our experiments. This is likely because the surgical videos used contain relatively clear procedural stages, reducing the need for explicit phase blending. Additionally, the MoG prior already models phase uncertainty, making the hard cluster assignment guided by $p(c|z, x)$ sufficient for the phase transitions present in our data.

We further assessed the method's robustness by testing different hyperparameters, specifically the ratio of features selected via DPP (1/4, 1/2, 3/4, and all features) from memories of size 4, 8, 16, and 24. Results in Fig. 5 (a-b) show that selecting half of the features performs best for most sizes. Performance without selection declines markedly with larger memories, suggesting the cross-attention module cannot fully utilize critical information from long sequences. We also studied the effects of cluster number in Fig. 5 (c-d). The performance did not improve with more clusters, peaking instead at 4 for the ESD dataset and 3 for the LC dataset. These optimal numbers align with the actual number of surgical phases present in the videos.

## 5. Discussion and Conclusion

In this paper, we propose a temporal memory enhancement method for surgical video semantic segmentation. The method comprises a local memory mechanism that dynamically selects diverse and relevant features via a greedy DPP and a global memory module based on CVAE-MoG to model phase-specific patterns. By integrating both local and global temporal contexts, our approach effectively handles the varying motion rates and structured workflow inherent in surgical videos. Extensive evaluations on the ESD and LC datasets show that our method outperforms state-of-the-art video segmentation methods, demonstrating its capability to manage complex temporal contexts in surgical video analysis.

While our method demonstrates robust performance through unsupervised, segmentation-driven phase clustering, it currently does not incorporate explicit phase labels that could provide additional supervisory signals. Phase labels, if available, could facilitate joint optimization of segmentation and phase recognition, potentially enhancing temporal consistency and semantic understanding. Future work could collect frame-accurate phase annotations

and explore a jointly optimized model that integrates phase classification with segmentation. Such a model could be compared with our current approach to quantitatively assess the trade-offs between label dependency and generalization capability, particularly in scenarios where phase boundaries are ambiguous or surgical techniques vary.

Our experiments were conducted on structured procedures such as ESD and CHO, which follow relatively linear workflows. However, many surgeries exhibit non-linear, adaptive, or interrupted workflows where phase transitions are less distinct. The unsupervised clustering strategy used in our global memory module learns inherent anatomical patterns without relying on predefined phase definitions, which may offer an advantage in such complex settings. Future work could involve evaluating our method on more diverse and challenging surgical videos, including those with non-sequential phases, multi-port interactions, and unexpected anatomical variations. This will further validate the robustness and generalizability of our temporal memory enhancement approach across a wider spectrum of surgical scenarios.

## 6. Acknowledgments

This work described in this paper was supported by a grant from the NSFC/RGC Joint Research Scheme sponsored by the Research Grants Council of the Hong Kong Special Administrative Region, China and the National Natural Science Foundation of China (Project No. N_CUHK410/23).

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

## Appendix A. Greedy Selection Algorithm

---

**Algorithm 1** Greedy Selection (Orthogonal Matching Pursuit)

---

**Initialize:** selected set: $\mathcal{B} = \emptyset$, weighted features: $v_t = r_t \cdot \bar{f}_{-t}$, kernel matrix: $K_{ij} = v_i^\top v_j$ for $i, j = 1, \ldots, T$

**For** $j = 1$ **to** $M$:

1. $t^* = \arg\max_{t \notin \mathcal{B}} K_{tt}$ (Select the item with the largest gain)

2. $\mathcal{B} \leftarrow \mathcal{B} \cup \{f_{-(t^*)}\}$

3. **If** $j < M$:
   $K \leftarrow K - \frac{1}{K_{t^* t^*}} K_{:,t^*} K_{:,t^*}^\top$     (Project onto orthogonal complement)

**Return** $\mathcal{B}$

---

## Appendix B. Derivation of the Evidence Lower Bound

### B.1. Derivation of Equation (4)

We start with the log-likelihood:

$$\log p(y|x) = \log \int \sum_{c=1}^{C} p(y, z, c|x) \, dz.$$

Introduce a variational distribution $q(z, c|x, y)$ (approximate posterior) over latent variables $z, c$

$$\log p(y|x) = \log \int \sum_{c=1}^{C} q(z, c|x, y) \cdot \frac{p(y, z, c|x)}{q(z, c|x, y)}.$$

This can be written as:

$$\log p(y|x) = \log \mathbb{E}_{q(z,c|x,y)} \left[ \frac{p(y, z, c|x)}{q(z, c|x, y)} \right].$$

By Jensen's inequality (since log is concave):

$$\log \mathbb{E}_{q(z,c|x,y)} \left[ \frac{p(y, z, c|x)}{q(z, c|x, y)} \right] \geq \mathbb{E}_{q(z,c|x,y)} \left[ \log \frac{p(y, z, c|x)}{q(z, c|x, y)} \right].$$

Therefore:

$$\log p(y|x) \geq \mathbb{E}_{q(z,c|x,y)} \left[ \log \frac{p(y|z, x) \, p(z, c|x)}{q(z, c|x, y)} \right].$$

Factorize the joint distribution $p(y, z, c|x) = p(y, z|x)p(z, c|x)$ and split the LHS:

$$\log p(y|x) \geq \mathbb{E}_{q(z,c|x,y)} \left[ \log p(y|z, x) \right] + \mathbb{E}_{q(z,c|x,y)} \left[ \log \frac{p(z, c|x)}{q(z, c|x, y)} \right].$$

Since the second term is negative KL divergence, we have

$$\log p(y|x) \geq \mathbb{E}_{q(z,c|x,y)} \left[ \log p(y|z, x) \right] - \mathcal{D}_{KL}[q(z, c|x, y) \| p(z, c|x)].$$

### B.2. Derivation of KL Divergence in Equation (8)

Start with the definition:

$$\mathcal{D}_{KL}[q(z,c|x,y)||p(z,c|x)] = \sum_{c=1}^{C} \int q(z,c|x,y) \log \frac{q(z,c|x,y)}{p(z,c|x)} \, dz..$$

Factorize the joint distributions:

$$= \sum_{c=1}^{C} \int q(z|x,y)q(c|x,y) \log \frac{q(z|x,y)q(c|x,y)}{p(z|c,x)p(c|x)} \, dz.$$

Substitute the defined distributions:

$$= \sum_{c=1}^{C} \pi_c^q \left[ \int \mathcal{N}(z|\mu_q, \sigma_q^2 I) \log \frac{\mathcal{N}(z|\mu_q, \sigma_q^2 I)}{\mathcal{N}(z|\mu_c, \sigma_c^2 I)} \, dz + \log \frac{\pi_c^q}{\pi_c^p} \right].$$

Final expression:

$$= \sum_{c=1}^{C} \pi_c^q \left( \log \frac{\sigma_c}{\sigma_c} + \frac{\sigma_c^2 + (\mu_c - \mu_c)^2}{2\sigma_c^2} - \log \pi_c^p \right).$$

## Appendix C. Evaluation Metrics

### C.1. Notations

For each class $k = 1, \ldots, K-1$, we define the predicted mask $P_i$ and the ground truth mask $T_i$ as two sets:

$$P_k = \{i \mid \hat{y}(i) = k, i \in I\}, \quad T_i = \{i \mid y(i) = K, i \in I\},$$

where $I$ denotes the set of pixels in the input image.

### C.2. wDice (weighted Dice)

The per-class Dice is derived based on cumulative masks:

$$\text{Dice}_k = \frac{2\,|P_k \cap T_k|}{|P_k| + |T_k| + \epsilon},$$

where $|\cdot|$ is pixel count. A small value $\epsilon$ is added for numerical stability. If $|P_k| + |T_k| = 0$, we set $\text{Dice}_k = 1$. Class weights are total positive pixels for class $k$:

$$w_k = \sum_{\text{samples}} |T_k|.$$

Then the weighted average (wDice) is:

$$\text{wDice} = \frac{\sum_k w_k \overline{\text{Dice}}_k}{\sum_k w_k}.$$

where $\overline{\text{Dice}}_k$ is the mean Dice over samples for class $k$.

## C.3. mIoU (Jaccard)

The IoU is calculated using the same cumulative masks:

$$\text{IoU}_k = \frac{|P_k \cap T_k|}{|P_k \cup T_k| + \epsilon}.$$

If $|P_k \cup T_k| = 0$, we set $\text{IoU}_k = 1$. The reported mIoU is the mean of per-class averages:

$$\text{mIoU} = \frac{1}{K-1} \sum_k \overline{\text{IoU}}_k$$

## C.4. F-score (boundary F1)

We compute boundaries of the binary masks and measures their overlap. Boundary extraction (conceptually):

$$B(P_k) = P_k \setminus \text{erode}(P_k), \quad B(T_k) = T_k \setminus \text{erode}(T_k).$$

Boundary precision/recall:

$$\text{Prec}_k = \frac{|B(P_k) \cap B(T_k)|}{|B(P_k)| + \epsilon}, \quad \text{Rec}_k = \frac{|B(P_k) \cap B(T_k)|}{|B(T_k)| + \epsilon}.$$

Boundary F-score:

$$\text{F1}_k = \frac{2\,\text{Prec}_k\,\text{Rec}_k}{\text{Prec}_k + \text{Rec}_k + \epsilon}.$$

If both boundaries are empty, $\text{F1}_k = 1$. The reported F-score is the mean of per-class averages:

$$\text{F1} = \frac{1}{K-1} \sum_k \overline{\text{F1}}_k.$$

