# OpenReview forum: "Temporal Memory Enhancement for Semantic Segmentation in Surgical Video"
_MIDL.io/2026/Conference — MIDL 2026 Poster_

### Official Review · Reviewer_qxSy · 2026-01-04

**Confidence:** 3
**Preliminary Rating:** 4
**Final Rating:** 5

**Summary:**

In this paper, the authors propose a novel method for enhancing temporal memory in surgical video semantic segmentation. This approach effectively overcomes the limitations of fixed-range local memories, which often struggle to capture the diverse dynamics of surgical motions and the extended phases of procedures. They introduce a local memory feature selection module that utilizes a greedy Determinantal Point Process to select a range of diverse and relevant past features while minimizing redundancy. In addition, a global memory module is developed based on a Conditional Variational Autoencoder with a Mixture of Gaussian to encode context specific to different phases of the surgery. Through extensive experiments on datasets from endoscopic submucosal dissection and laparoscopic cholecystectomy, the method demonstrates state-of-the-art performance, effectively handling the complex temporal dependencies inherent in surgical workflows.

**Strengths:**

Utilizing DPP for feature selection effectively addresses the varying movement rates of surgeons, thereby preventing memory overload caused by redundant data during slower moments. Visual consistencies are captured across different surgical phases, offering a high-level semantic context that traditional recency-based memory systems frequently overlook. The framework of temporal memory enhancement method for video semantic segmentation described in Figure 2 is excellent and helps a reader grasp the concepts better. This paper in general delivers value to the community by proposing a useful solution for segmenting small, critical structures within dynamic environments.

**Weaknesses:**

1. Could the experimental results perhaps have included other datasets as well to provide more comprehensive evaluation?
2. Why is the F-score in general lower on the CHO dataset as compared to the ESD dataset?
3. Are there any other experimental metrics you could have considered for evaluation purposes?

**Detailed Comments:**

The comments are the same as in the weaknesses section above.

**Justification Of Final Rating:**

I'll maintain my rating for an accept after the thoughtful comments by the authors. They addressed all the points posed earlier in the initial review, especially on the consideration of additional datasets for experimentation.

**Justification Of The Preliminary Rating:**

This paper in general delivers good value to the community by proposing a novel solution for segmenting small, critical structures within dynamic environments, and is written in a nice, fluid style that makes the concepts easy to grasp even for a lay reader.

**Questions To Address In The Rebuttal:**

It would be useful to expand a bit more on the choice of experiment settings described in page 8 of the paper.

---

> ### Author Response · Authors · 2026-01-24
>
> **Comment 1:** Could the experimental results perhaps have included other datasets as well to provide more comprehensive evaluation?
>
> Thank you for the suggestion regarding dataset diversity. We fully agree that evaluating on additional surgical procedures would strengthen the generalizability of our findings. However, constructing high-quality surgical video segmentation datasets is challenging due to the need for precise, pixel-level annotations of anatomical structures by clinical experts, which is a time-intensive and costly process.
>
> In this work, we strategically selected two established benchmarks, submucosal dissection and laparoscopic cholecystectomy, to validate our method. These datasets represent fundamentally different surgical domains with distinct visual textures, motion patterns, and segmentation targets. Demonstrating improved performance across these two challenging and diverse scenarios provides strong initial evidence for our method's robustness.
>
> We are actively expanding this line of research and plan to include datasets from other common procedures, such as laparoscopic prostatectomy and retinal microsurgery, in future work.
>
> **Comment 2:** Why is the F-score in general lower on the CHO dataset as compared to the ESD dataset?
>
> Thank you for raising this important question regarding the lower Boundary F-score on the CHO dataset compared to the ESD dataset. Our reported F-score is the *Boundary F-score (F1)*, a metric highly sensitive to contour alignment. The F1 score of each class $i$ is calculated as:
>
> $$
> \mathrm{F1}_i = \frac{2\,\mathrm{Prec}_i\,\mathrm{Rec}_i}{\mathrm{Prec}_i + \mathrm{Rec}_i + \epsilon},\quad
> \mathrm{Prec}_i = \frac{|B(P_i) \cap B(T_i)|}{|B(P_i)| + \epsilon},\quad
> \mathrm{Rec}_i = \frac{|B(P_i) \cap B(T_i)|}{|B(T_i)| + \epsilon},
> $$
>
> where $B(P_i)$ and $B(T_i)$ denote the predicted and target boundaries of class $i$. This metric directly penalizes boundary fragmentation or displacement. The lower score on CHO could arise from three dataset-inherent challenges:
>
> **1. Smaller anatomical structures.** Fine structures like the cystic artery can be exceptionally narrow (sometimes as small as 1 mm in diameter). In CHO dataset, this translates to a very small number of boundary pixels. A tiny displacement could lead to a sharp reduction in both precision and recall. In ESD, structures are larger, and the same absolute error has a much smaller impact.
>
> **2. Obscure and ambiguous boundaries.** Laparoscopic cholecystectomy (CHO) involves frequent tissue deformation, motion blur, and fluid occlusion, which blur and obscure anatomical contours. This boundary ambiguity makes it difficult to achieve precise pixel‑wise alignment, directly lowering the Boundary F-score. In contrast, ESD scenes offer more stable, clearly defined tissue layers.
>
> **3. Higher classification complexity.** CHO has 6 classes to distinguish versus 3 in ESD. This finer classification task increases inter-class confusion. Consequently, a boundary pixel error often creates a simultaneous false positive and false negative, which is doubly penalized in boundary-focused metrics.
>
> To support this analysis, we have added per-class boundary F-score and Dice results in the revised manuscript (Fig. 3). They confirm that all methods score lower on CHO’s small and boundary-ambiguous structures, validating that the performance difference stems from dataset characteristics rather than methodological limitations. Detailed information for all evaluation metrics has been included in the appendix to facilitate a clearer understanding of our results.
>
>
>
> **Comment 3:** Are there any other experimental metrics you could have considered for evaluation purposes?
>
> Thank you for raising this important point regarding evaluation metrics. To provide a more granular assessment of segmentation performance, we have incorporated per-class Dice scores and F-scores in our analysis. These metrics are especially valuable for isolating performance on small, critical anatomical structures—such as the cystic duct and blood vessels—where the precise localization are paramount for downstream clinical tasks, including the identification for critical view of safety and procedural guidance.
>
> We have added the additional results in Fig. 3 of the revised manuscript. The results show that while the performance benefits on larger structures like the gallbladder or submucosal tissue are limited, our approach achieves significant improvement on critical, small-scale anatomy compared to prior video segmentation baselines. This demonstrates that our memory enhancement strategy offers particular practical value where precise segmentation is most safety-critical.

---

> > ### Comment · Reviewer_qxSy · 2026-01-26
> >
> > Thanks for the response. The authors addressed my questions. I am keeping my score as an accept.

---

### Official Review · Reviewer_oVKh · 2026-01-07

**Confidence:** 5
**Preliminary Rating:** 4
**Final Rating:** 4

**Summary:**

This paper proposed a memory enhancement method for surgical video semantic segmentation, which helps both short and long-term temporal modeling. A DPP-based feature selection is used to reduce redundancy and the global phase part captures common patterns within each surgical phase. This method outperforms existing approaches on ESD and CHO datasets.

**Strengths:**

1. Combines local and global temporal information through effective memory enhancement designs.

2. Accounts for the unique nature of surgical videos, which is critical for accurate segmentation.

3. Outperforms multiple baselines with consistent results.

**Weaknesses:**

The global memory relies on the assumption of phase consistency. However, in real scenarios, phase boundaries are often unclear, making it difficult to distinguish different stages. In addition, variations in surgical techniques across surgeons may further reduce the reliability of phase-based modeling.

**Detailed Comments:**

Robustness to ambiguous phase boundaries could be improved by introducing phase perturbation during training if possible.

**Justification Of Final Rating:**

Thanks for your response! And you already add the experiment table for this stage. I think you already solve all my questions and I am happy to accept this paper. I am looking forward to your final version of this paper. Good luck.

**Justification Of The Preliminary Rating:**

My rating is based on the method design (both short and long temporal modeling) and the experiment results on two public datasets. Although real-world scenarios are more complex, the proposed method demonstrates clear novelty.

**Questions To Address In The Rebuttal:**

No.

---

> ### Author Response · Authors · 2026-01-24
>
> **Comments:** (1) The global memory relies on the assumption of phase consistency. However, in real scenarios, phase boundaries are often unclear, making it difficult to distinguish different stages. (2) Variations in surgical techniques across surgeons may further reduce the reliability of phase-based modeling. (3) Robustness to ambiguous phase boundaries could be improved by introducing phase perturbation during training if possible.
>
> Thank you for your insightful comment regarding phase consistency and surgical technique variation. Our method is specifically designed to address these challenges.
>
> Our approach *does not rely on predefined phase consistency or discrete labels*. Instead, it uses unsupervised clustering based on a Mixture of Gaussians (MoG) distribution to model the inherent uncertainty between phases. This allows the model to handle ambiguous, gradual transitions effectively, as frames are grouped by visual and anatomical similarity from segmentation masks rather than by hard phase boundaries.
>
> Furthermore, by deriving context directly from segmentation masks—which reflect underlying anatomy—our method bypasses dependence on potentially inconsistent human-defined phase labels that vary across surgeons and institutions. This focuses the model on structural commonalities, enhancing its robustness to technical variations.
>
> To explore robustness to phase ambiguity, we implemented an alternative *Soft-Mix* strategy that generates the global context $\tilde{z}^l$ as a weighted average of samples from all Gaussian components:
>
>
> $$\tilde{z}^l = \sum_{c=1}^C \pi_c \tilde{z}^l_c, \quad \tilde{z}^l_c \sim p(z \mid c, x),$$
>
> where $\pi_c$ is the mixing probability. This introduces a form of phase perturbation by blending contexts from multiple clusters. Based on our experiment results on the ESD and CHO datasets, the Soft-Mix variant did not outperform the original hard-assigned cluster approach. This could be attributed to two factors: (1) the surgical videos used in our study contain relatively well-defined procedural stages. The inherent uncertainty between phases may not be sufficiently high to benefit from explicit blending; (2) the MoG prior already captures uncertainty in phase assignment through its probabilistic formulation. The hard assignment, guided by the posterior $q(c \mid x, y)$, appears sufficient to model the phase transitions present in our data.
>
> | Methods  | LC mIoU (%) | LC F-score (%) | LC wDice (%) | ESD mIoU (%) | ESD F-score (%) | ESD wDice (%) | FPS  |
> |----------|-------------|----------------|--------------|--------------|-----------------|---------------|------|
> | Soft-Mix | 86.2±0.85   | 47.6±2.91      | 92.2±0.78    | 86.2±1.12    | 75.2±1.87       | 89.5±0.54     | 66.2 |
> | Ours     | 86.6±0.83   | 47.8±3.21      | 92.7±0.38    | 86.9±1.13    | 76.4±2.12       | 90.5±0.59     | 66.3 |
>
> Nevertheless, we believe Soft-Mix remains a valuable direction for more complex or heterogeneous surgical workflows, and we plan to investigate it further in future work. We have included the additional results and discussion in our revised manuscript.

---

### Official Review · Reviewer_zv86 · 2026-01-08

**Confidence:** 5
**Preliminary Rating:** 5
**Final Rating:** 5

**Summary:**

The authors propose a temporal memory enhancement method to enrich the local context and global surgical phase context for surgical video semantic segmentation. The method is evaluated on endoscopic submucosal dissection and laparoscopic cholecystectomy datasets, where the authors report state‑of‑the‑art performance.

**Strengths:**

(1) The idea of enriching the local temporal context by incorporating the global phase context for surgical video semantic segmentation seems novel.

(2) The authors present a thorough literature review, highlighting the novelty of their design.

(3) The authors conduct several ablation studies to justify their architectural choices and evaluate the method across multiple datasets.

**Weaknesses:**

(1) When comparing the proposed method with other state‑of‑the‑art approaches in Table 1, the backbones seem to be different. If the pre-training data and backbones are different, it becomes difficult to attribute performance gains to the proposed contribution rather than to a stronger backbone or more favorable pre-training.

(2) Please conduct a grammar and typo check for the paper.

(3) Please conduct a reference check and update the reference if needed.

Optional to address: (4) The use of phase context proposed by the authors seems to be a great idea. How does it compare to directly leveraging phase labels through a joint optimization of segmentation and phase classification?

Optional to address: (5) The test datasets contain only 3–4 surgical phases, and the phase progression appears largely linear. For future work, it would be valuable to evaluate the method on more complex surgical procedures that involve a larger number of phases and non‑linear phase transitions.

**Detailed Comments:**

(2) Please conduct a grammar and typo check for the paper:

(a) “which has achieved excellent performance ” to “which have achieved excellent performance”.

(b) “provide the long-term phase context” to “provides the long-term phase context”.

(c)  “laparoscopic cholecystectomy (CHO)” or “laparoscopic cholecystectomy (LC)”?

(3) Please conduct a reference check and update the reference if needed.

For example, the paper titled “Semantic video segmentation by gated recurrent flow propagation” has been published.

**Justification Of Final Rating:**

I would like to thank the authors for their response. The revision offers clear clarification of the experiments and effectively resolves the grammatical and reference issues previously noted. The inclusion of additional potential future work further strengthens the manuscript. In my view, the overall quality of the paper is very strong. I will maintain my rating of strong accept and recommend the paper for an oral presentation and consideration for a special issue.

**Justification Of The Preliminary Rating:**

From my perspective, this is primarily a methodology paper, and the proposed approach appears novel. The literature review, method presentation, and experimental evaluation are all well‑executed, and the overall quality of the manuscript is strong. I vote for strong accept for this paper.

**Questions To Address In The Rebuttal:**

Please prioritize addressing weaknesses (1), (2), and (3). Addressing weaknesses (4) and (5) is optional.

---

> ### Author Response · Authors · 2026-01-24
>
> **Response to weakness 1:**
>
> Thank you for raising this important point regarding backbone consistency. We agree that fair attribution is crucial. We would like to clarify that the performance gains reported in Table 1 are indeed attributable to our proposed memory modules, based on the following two key aspects of our experimental design:
>
> **1. Controlled Comparison Within the Same Backbone Family:** In Table 1, all methods built upon the SAM architecture—namely SAM2, MemSAM, SurgSAM2, and our method—use the identical Hiera-Large encoder pre-trained on the SAM2 dataset. They were also fine-tuned using the same adapter-based protocol on our surgical video data. Therefore, the superior performance of our method over these specific baselines can be directly attributed to our temporal enhancement strategy, not to a stronger or different backbone. The inclusion of models with different backbones (Mask2Former, FRGM) serves primarily as a performance baseline and demonstrate the relative strength of the SAM2-based architecture against other strong video and surgical segmentation models.
>
> **2. Direct Evidence from Ablation Studies:** To isolate the contribution of our proposed components, we conducted systematic ablation studies (Table 2). The results show that removing the diversity or relevance scoring in our local memory selection causes a clear performance drop; Simplifying the global phase context modeling (e.g., removing clustering or using a simpler method like K-means) also leads to inferior results. These controlled experiments, where only our proposed modules are altered while the backbone remains fixed, confirm that the observed gains stem from our architectural innovations in memory enhancement.
>
> We have added the explanations about the backbone in Section 4.2 to clarify the potential confounding effect. We hope this clarification could strengthen our core conclusions of the proposed memory enhancement strategy.
>
> **Response to weakness 2 & 3**
>
> Thank you for your meticulous reading and valuable feedback. We have carefully checked the manuscript and implemented all the corrections you suggested regarding grammar, terminology, and references. The changes have been integrated into the revised manuscript.
>
> **Response to weakness 4**
>
> Thank you for highlighting the potential of leveraging phase labels through joint optimization, which is a valuable direction for future work. Our current study focuses on an unsupervised, segmentation-driven approach to derive phase context, which could offer the following potential robustness advantages in real-world applications:
>
> **1. Robustness to Ambiguous Phase Boundaries:** Surgical phases often transition gradually with ambiguous boundaries. By clustering frames based on visual and anatomical similarity from segmentation masks, our method adapts better to ambiguous transitions than models reliant on discrete phase labels.
>
> **2. Independence from Surgical Technique Variation:** Phase definitions could vary across surgeons and institutions. Learning from segmentation masks allows our model to capture underlying structural commonalities, improving generalizability over human-defined labels.
>
> **3. Addressing the Label Scarcity Challenge:** As noted, high-quality, frame-accurate phase labels are scarce and expensive to obtain. Our approach provides a way to utilize abundant segmentation annotations to implicitly inform the temporal model, which is a significant practical benefit.
>
> We agree that a direct comparison with a supervised phase-classification joint model would be insightful. We will collect phase annotations and conduct this comparative analysis as an important future work, building upon the unsupervised foundation laid by this study. Furthermore, we have added a paragraph to the Discussion (Section 5) section to acknowledge this point.
>
> **Response to weakness 5**
>
> Thank you for the insightful suggestion. We fully agree that evaluating our method on more complex surgical workflows represents a crucial next step to demonstrate its robustness and generalizability. By using unsupervised clustering strategies, our method learns inherent anatomical patterns, rather than relying on predefined phase labels, making it particularly suitable for segmentation in surgical videos with non-linear workflows. In future work, we will collect data from such complex surgeries to quantitatively demonstrate this robustness compared to phase label-dependent models. We have acknowledged this direction in the discussion section.

---

### Author Rebuttal · Authors · 2026-01-24

**Rebuttal:**

We appreciate the hard work and constructive suggestions from chairs and all reviewers.  Please kindly refer to the point-to-point response for our rebuttal. And we provide our revised manuscript with highlights here.

**Supporting Material:**

/attachment/e2ab192526232faef30dd8b2d1f1bc555e620c8b.pdf

---

### Meta-Review · Area_Chair_nBur · 2026-02-03

**Recommendation:** Accept (Oral)
**Confidence:** 4

**Metareview:**

The reviewers provided positive ratings, minor concerns. The author's rebuttal demonstratesthe strong merit of the paper's impact, and strengthens the paper's contribution. As all evidence stand, the paper can be accepted.

---

### Decision · Program_Chairs · 2026-02-13

Accept (Poster)